# Clinical management and genomic profiling of pediatric low-grade gliomas in Saudi Arabia

**Nahla A. Mobark[1]**☯**, Musa Alharbi[1]**☯**, Lamees Alhabeeb[2], Latifa AlMubarak[2], Rasha Alaljelaify[2], Mariam AlSaeed[2], Amal Almutairi[2], Fatmah Alqubaishi[2], Maqsood Ahmad[3], Ayman Al-Banyan[3], Fahad E. Alotabi[3], Duna Barakeh[4], Malak AlZahrani[4], Hisham Al-Khalidi[4], Abdulrazag Ajlan[4], Lori A. Ramkissoon[5], Shakti H. Ramkissoon[6,7], Malak Abedalthagafi**[2]*

**1** Department of Paediatric Oncology Comprehensive Cancer Centre, King Fahad Medical City, Riyadh, Kingdom of Saudi Arabia, **2** Genomics Research Department, Saudi Human Genome Project, King Fahad Medical City and King Abdulaziz City for Science and Technology, Riyadh, Saudi Arabia, **3** Department of Neuroscience, King Fahad Medical City, Riyadh, Kingdom of Saudi Arabia, **4** Department of Pathology, King Khalid Hospital, King Saud University, Riyadh, Saudi Arabia, **5** Department of Neurosurgery, University of North Carolina School of Medicine, Chapel Hill, NC, United States of America, **6** Wake Forest Comprehensive Cancer Center and Department of Pathology, Wake Forest School of Medicine, Winston-Salem, NC, United States of America, **7** Foundation Medicine Inc., Morrisville, NC, United States of America

☯ These authors contributed equally to this work.
* malthagafi@kfmc.med.sa

**Data Availability Statement:** All relevant data are within the paper and its Supporting Information files.

## Abstract

Pediatric Low Grade Gliomas (PLGGs) display heterogeneity regarding morphology, genomic drivers and clinical outcomes. The treatment modality dictates the outcome and optimizing patient management can be challenging. In this study, we profiled a targeted panel of cancer-related genes in 37 Saudi Arabian patients with pLGGs to identify genetic abnormalities that can inform prognostic and therapeutic decision-making. We detected genetic alterations (GAs) in 97% (36/37) of cases, averaging 2.51 single nucleotide variations (SNVs) and 0.91 gene fusions per patient. The *KIAA1549-BRAF* fusion was the most common alteration (21/37 patients) followed by *AFAP1-NTRK2* (2/37) and *TBLXR-PI3KCA* (2/37) fusions that were observed at much lower frequencies. The most frequently mutated) genes were *NOTCH1-3* (7/37), *ATM* (4/37), *RAD51C* (3/37), *RNF43* (3/37), *SLX4* (3/37) and *NF1* (3/37). Interestingly, we identified a *GOPC-ROS1* fusion in an 8-year-old patient whose tumor lacked *BRAF* alterations and histologically classified as low grade glioma. The patient underwent gross total resection (GTR). The patient is currently disease free. To our knowledge this is the first report of *GOPC-ROS1* fusion in PLGG. Taken together, we reveal the genetic characteristics of pLGG patients can enhance diagnostics and therapeutic decisions. In addition, we identified a GOPC-*ROS1* fusion that may be a biomarker for pLGG.

## Introduction

Gliomas are common tumors in children and adolescents that display a broad range of clinical behaviors.[1] Most pediatric gliomas are benign and slow-growing lesions classified as grade I

**Funding:** This Study was funded by KFMC-IRF 17-65 (MA) and Sanad Cancer research foundation RGP 2017-1 (MA). The funders had no role in study design, data collection and analysis, decision to publish, or preparation of the manuscript.

**Competing interests:** One of our authors is affiliated with Foundation Medicine Inc. The author only contributed relevant pathology consultation. This affiliation does not alter our adherence to PLOS ONE policies on sharing data and materials.

or II by the World Health Organization (WHO) classification criteria.[2–8] The most recent WHO classification in 2016 describes their histological features and provides a grading or malignancy scale.[6] Pediatric low-grade gliomas (pLGGs) account for approximately 35% of all childhood brain tumors and differ from adult low-grade gliomas (aLGGs) as they are seldom associated with *IDH1/2* mutations, rarely undergo malignant transformation, and display high survival rates in response to traditional therapy.[9–12]

The most common pLGG tumors in children are pilocytic astrocytoma (PA, Grade I) and diffuse astrocytoma (Grade II).[2–5] Other less common tumor types include pilomyxoid astrocytoma (Grade II), pleomorphic xanthoastrocytoma (PXA, Grade II), ganglioglioma (Grade I), angiocentric glioma (Grade I), subependymal giant cell astrocytoma (Grade I), and oligodendroglioma (Grade II).[6,13,14] pLGGs can be difficult to classify as they occur throughout the central nervous system (CNS) and often demonstrate overlapping microscopic features.[15,16] Historically, the cerebellum is the most common location with cerebellar LGGs accounting for 15% to 25% of all pediatric CNS tumors. These are followed by hemispheric (cerebral) gliomas (10%-15%), gliomas of the deep midline structures (10%-15%), optic pathway gliomas (5%), and brainstem gliomas (2%-4%).

Surgery is the primary therapeutic modality for pilocytic astrocytomas and other LGGs.[17–21] Gross total resection (GTR) is often curative, despite the presence of residual microscopic disease. Clinical management strategies of children with subtotal resections are typically developed at a multidisciplinary pediatric neuro-oncology tumor board on a case-by-case basis. If the likelihood of functional impairment is minimal and neurosurgical intervention is deemed feasible, repeat surgery can often remove the residual tumor. In most cases, a "wait and see approach" is advocated, with follow-up brain MRI performed at 3 to 6 month intervals. Because pLGG tumors tend to be indolent by nature, the decision for repeat resection or adjuvant treatment can be postponed until measurable progression evidenced through neuroimaging or clinical symptoms are observed.[22–25] This interval may last several years and some tumors never progress.[13,26]

When GTR is not possible, front-line chemotherapy regimens per the Children's Oncology Group Protocol A9952 are advocated. This involves a combination of carboplatin and vincristine that provides stable disease and tumor regression for an extended period. Chemotherapy also permits improved surgical resection of previously unresectable lesions.[27–32] Combined carboplatin and vincristine results in tumor reduction or stable disease and a 3-year PFS of 68%.[2,27,33–35] Radiotherapy is typically contraindicated in children, particularly those with *NF1* germline mutations, PAs, and other LGGs, including cases of diencephalic and optic pathway tumors. Even highly focused radiation therapy at these locations does not eliminate the associated cognitive, endocrine, or vascular risks.[33,36,37]

In this study, we sought to illustrate how correlating genetic alterations with histologic and clinical features can improve pLGG classification and treatment decisions for patients in Saudi Arabia (SA). Our cohort included cases from a tertiary care center in King Fahad Medical City (KFMC), which is a primary referral center for pediatric neoplasms and King Khalid University Hospital (KKUH) in SA and reflects the distribution of pLGG subtypes across the kingdom.

## Materials and methods

### Patient cohort

This retrospective study was performed with IRB#16–310 following the relevant ethical guidelines and regulations from the King Fahad Medical City, Riyadh, KSA. King Fahad Medical City's International Review Board reviewed and approved this study before the study began

and waived the requirement for informed consent for the archival samples. The study was performed on 37 children where tissue was available (age <16 years) who were newly diagnosed with pathologically confirmed pLGG between January 2011 and January 2017. We reviewed the molecular, clinical and therapeutic aspects and treatment outcomes of the pLGG patients in KFMC (S1 Fig). We collected essential demographic and disease-specific characteristics and radiology images to assess the extent of tumor resection. Information on neurosurgical management was obtained from operative records and standardized neurosurgical reports. All data were fully anonymized before we accessed them. Archived pathology specimens were reviewed by a board-certified neuropathologist (MA). Progression-free survival (PFS) was defined as the minimum time to tumor progression, second malignancy, or death from any cause.

### Next-generation sequencing

Next-generation sequencing (NGS) was performed using the Oncomine Comprehensive Assay v3 system, a targeted assay that enables the detection of relevant SNVs, CNVs, gene fusions, and indels from 161 genes (S2 Fig). Multiplex DNA primers were used to prepare amplicon libraries from formalin-fixed paraffin-embedded (FFPE) tumor samples. Assays were performed using the Ion S5 System and Ion 540 Chip (Thermofisher Scientific, USA).

### Statistical analysis

Overall survival (OS) was measured from the date of diagnosis until the date of death from any cause or date of last contact. PFS and OS were estimated using the Kaplan-Meier method. A P-value < 0.05 was considered statistically significant. A secure electronic database was created for storage and data analysis. The data were entered and analyzed using SPSS statistical package version 23.

## Results

### Cohort demographics and clinical management

Thirty-seven patients were assessed (19 males, 18 females) with a median age at diagnosis of 1–12 years (range: 12–154 months) with histologically proven low-grade astrocytoma (Grades I, II) (Fig 1A). Most tumors (31/37 cases, 83.7%) were classified as pilocytic astrocytomas. Four patients were classified with pleomorphic xanthoastrocytoma (10.8%) and two patients had diffuse gliomas (5.4%), (Fig 1B).

Cerebellar tumors were encountered in 18/37 patients (48.6%) accounted for 8/37 tumors (21.6%). Cerebral hemisphere/cortex tumors also occurred in 8/37 patients (21.6%), the second most common sites. A total of 4/37 patients had hypothalamic tumors with optic pathway involvement, 3/37 patients (8.10%) had suprasellar masses, 2/37 (5.4%) patients had spinal cord tumors as the primary site, and 2 (5.40%) patients had brain stem tumors (S1 Table and Fig 1C).

Amongst the patients, 26/37 (70.3%) had initial surgery followed by observational serial MRI (Fig 2A) and 9 of these patients experienced relapse/progression (Fig 2B). In total, 10 patients received adjuvant chemotherapy after surgery, two of which had relapse/progression after first-line chemotherapy, with a single patient relapsing on two occasions (S1 Fig). A single patient (1/37) received adjuvant radiotherapy due to incomplete surgical resection of the tumor and showed no relapse (Fig 2A). Amongst the surgical procedures employed, 10/26 underwent complete surgical excision, 7/26 underwent partial excision, 2/37 had subtotal resection, and a single patient underwent biopsy. The surgical procedures for the remaining patients were not defined. All the patients are still alive, whilst 11 (29.7%) experienced relapse/progression. The median progression-free survival time was 36.5 months.

**A**

| Characteristics | N= |
|---|---|
| **Gender** | |
| Males | 19 |
| Females | 18 |
| **Age at Diagnosis** | |
| Mean | 6.8 (yrs) |
| Range | 1-12.8 (yrs) |

**B**

| Histology | N= |
|---|---|
| Pilocytic Astrocytoma | 31 |
| Pleomorphic Xanthoastrocytoma | 4 |
| Difuse Glioma | 2 |

**C**

| Location | N = |
|---|---|
| Cerebellum | 18 |
| Cerebral cortex and hemisphere | 8 |
| Optic/hypothalamus | 4 |
| Suprasellar mass | 3 |
| Brain stem | 2 |
| Spine | 2 |

**Fig 1. Analysis of the SA LGG cohort.** (a) Patient demographics, (b) pLGG tumor histology, and (c) tumor locations.

### Genetic alterations in pLGGs

Gene alterations were identified in 36/37 (97.3%) of pLGGs, averaging 2.51 single nucleotide variations and 0.91 gene fusions per patient (Fig 3A). The *KIAA1549-BRAF* fusion was most common (21/37 patients) followed by *AFAP1-NTRK2* (2/37) and *TBLXR-PI3KCA* (2/37)

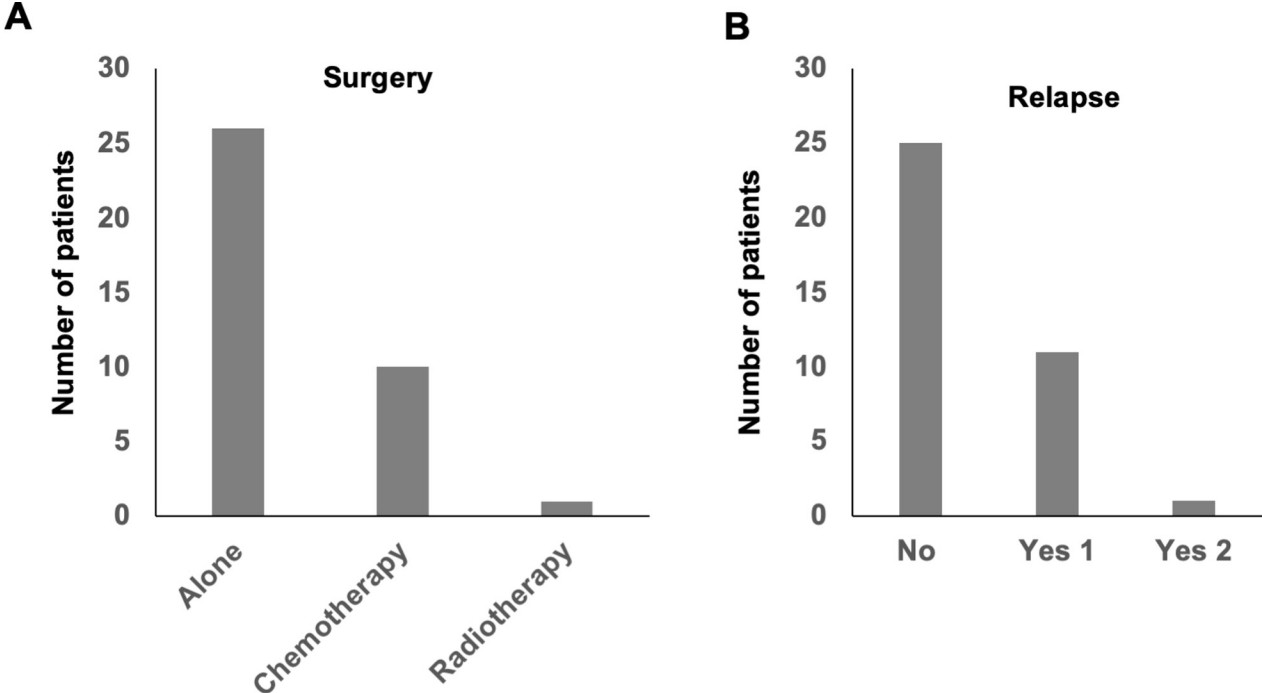

**Fig 2. pLGG treatment and recurrence.** Bar graphs show the treatment modalities (a) and tumor recurrence (b) employed in the SA pLGG cohort.

fusions. The most commonly mutated genes were *NOTCH13* (7/37), *ATM* (4/37), *RAD51C* (3/37), *RNF43* (3/37), *SLX4* (3/37) and *NF1* (3/37). Less frequently altered (observed in less than 3 patients each) genes included *FANCD2, FGFR1, BRCA2, CDC4, KIF5B, RET, AKAP, SLX4, MSH6, NTRK, CCDC170, MLH1, AGK, PDGFRB, EIF3B, FGFR2, CDKN2A/B, PTCH1, SETD2, SLX4, MSH6 NTRK* and *CDK4* (summarized in S2 Fig). Co-occurring alterations were detected in 19/37 patients with *BRAF* fusions (S2 Table). Two of these patients harbored both *ESR1-CCDC170* and *KIAA1549-BRAF* fusions, whilst a single case co-harbored *AGK-BRAF* and *KIAA1549-BRAF* fusions. One case harbored *TBL1XR1-PIK3CA, EIF3E-RSPO2* fusions (S2 Table). Regarding anatomic location and *BRAF* fusions, most were located in cerebellum/posterior fossa (S1 and S2 Tables). Of the relapse cases, 10/12 harbored *KIAA1549-BRAF* fusions. Of the cases lacking *KIAA1549-BRAF* fusions, a patient harboring a *GOPC-ROS1* fusion of was of interest, as this alteration has been previously reported in an undefined glioblastoma patient. The clinical outcome of this case was not previously reported in the literature.[38,39]

## Identification of the GOPC-ROS1 fusion in a single LGG patient

Golgi-associated PDZ and coiled-coil motif-containing (GOPC) protein regulates the intracellular trafficking of membrane proteins.[40] The ROS proto-oncogene 1 is a receptor tyrosine kinase expressed in lung and brain tissues.[41] *GOPC-ROS1* fusions have been identified in a "not otherwise specified" (NOS) single case of glioblastoma (deemed to be neither pLGG nor pHGG) in which the tumor also harbored mutations in other glioma-associated genes, including *TP53* and *PTPN11*,[38] and in one case of pHGG.[39] In the *GOPC-ROS1* fusion pHGG case, the patient underwent gross total resection, at 4 years of age, followed by adjuvant high-dose chemotherapy and autologous hematopoietic stem cell rescue. At 30 months post-transplantation, the patient remained disease-free.[39]

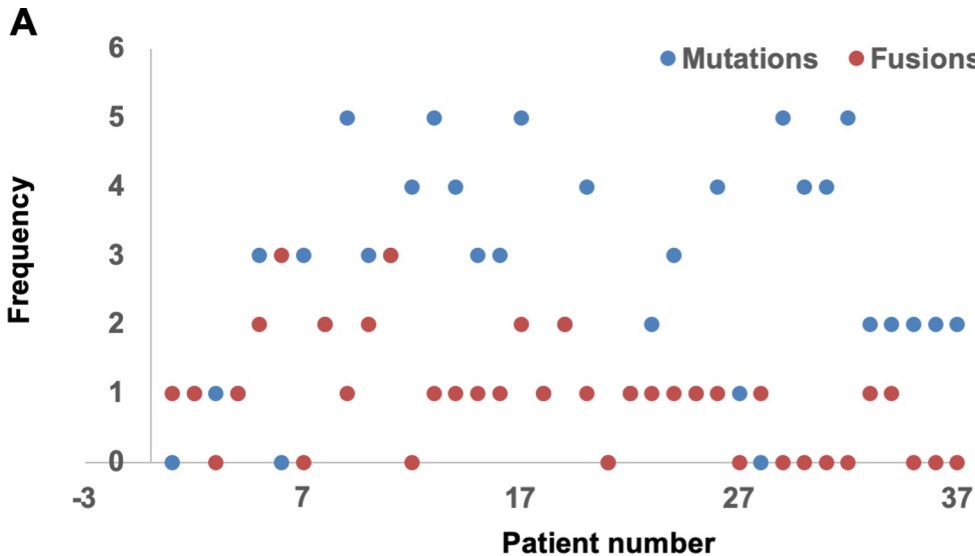

**Fig 3. Genetic landscape of the pLGG tumors.** (a) Number of genetic mutations and gene fusions observed across the cohort, (b) gene fusions observed in the pLGG tumors, (c) Mutational burden of the LGG tumors.

We detected a likely pathogenic *GOPC-ROS1* fusion in a pLGG patient who also harbored a *RAD15C* variant of uncertain significance. He is a previously healthy 8-year-old boy who presented with unprovoked recurrent convulsions. MRI imaging revealed a left parietal mass measuring (6 × 5 cm) with a mass effect and vasogenic edema (Fig 4). The patient underwent GTR. The patient is currently disease free and showed excellent postoperative recovery with no neurological deficits or evidence of progression (S1 Table). Histopathological examination revealed a low-grade glioma with low-to-moderate cellularity and a biphasic growth pattern comprising tumor cells with piloid and oligodendroglial morphologies and associated with Rosenthal fibers (Fig 4) Postoperative brain MRI revealed a gross total resection with no residual tumor identified (Fig 4). Spine MRI was unremarkable with no spinal seeding metastases. No adjuvant chemotherapy or radiation therapy was indicated (S1 Table). Currently, the patient is stable with no symptoms or signs suggestive of tumor recurrence, managed with stable serial follow-up MRI.

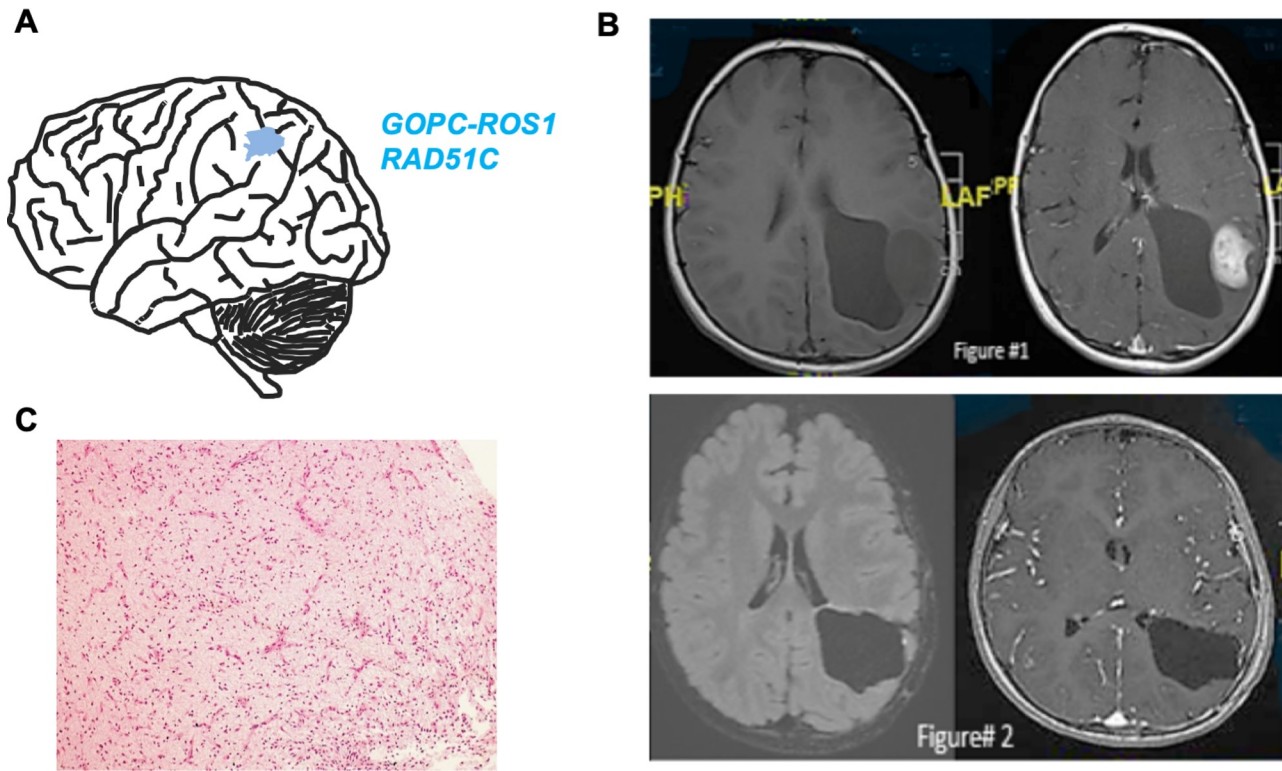

**Fig 4. GOPC-ROS1 fusion and disease pathology.** (a) Schematic of the tumor location and co-occurring mutations, (b-c) Postoperative MRI showing gross total surgical resection of the mass with no residual tumor identified. (d) Photomicrograph of an H&E-stained tumor section, demonstrating findings of low-grade glioma with a low to moderate cellularity, biphasic architecture, piloid and oligodendroglial tumor cells and Rosenthal fibers.

## Discussion

In this study, we reviewed the clinical management and performed a targeted genetic screening of a panel of cancer-related genes in 37 pediatric LGGs. Amongst the pLGGs, the most common alterations were *KIAA1549-BRAF* fusions (26/37). From histological analysis, most patients were diagnosed with pilocytic astrocytomas (31/37). This was comparable to previous findings in which 60% to 80% of PAs harbored *BRAF* fusions.[42] *BRAF* V600E mutations were observed in only 2/37 patients, while *H3F3A* (K27M) histone mutations were not detected. Tumors were predominantly located in the cerebellum/posterior fossa (18/37) and cerebral hemisphere/cortex (8/37). The majority of tumors harboring *BRAF* fusions were located in the cerebellum/posterior fossa (Fig 1C, S2 Table), which was consistent with the association between tumor location and *BRAF* fusions reported in other cohorts.[42,43] Tumors in the cerebellum are traditionally amenable to surgical intervention, with resection rates of ~94% and an overall tumor recurrence rate of ~19%.[42] We observed higher rates of recurrence (12 of 37 patients, 32.43%), raising the possibility that *BRAF* fusions may not be directly associated with an improved outcome in SA pLGGs, as reported in previous studies. [43,44] Because *KIAA1549-BRAF* is rarely detected in pHGGs, including anaplastic astrocytoma and glioblastoma, *BRAF* fusions can genetically distinguish pLGG from pHGG in the Saudi cohort. We suggest that in the future, the identification of *BRAF* fusions can guide patient treatment as targeted molecular therapies are discovered. It is important that a diagnostic test for these fusions is readily available in SA in a clinical setting.

Regarding gene variants, mutations in the Notch genes were most frequent (Fig 3C, S2 Table). Notch signaling is evolutionarily conserved and known to regulate cell proliferation, apoptosis, migration, and differentiation. In mammals, Notch signaling is composed of Notch1–4 receptors and Dll1-3–4, Jagged1–2 ligands that develop and maintain the CNS. The frequency of these mutations is perhaps surprising since higher expression of ASCL1, Dll1, Notch1, -3, -4 have been shown to correlate with a higher glioma grade and poorer prognosis, implicating Notch signaling in more undifferentiated and aggressive tumor phenotypes. We observed no association between Notch mutants and relapse/disease progression in our cohort, indeed a patient with co-occurring Notch2/3 mutations did not relapse following surgical intervention (S2 Table).

Among the less frequent mutations, we observed alterations in *RAD51C*, a component of the DNA double-strand repair pathway, the E3 ubiquitin ligase RNF43, and the central checkpoint gene *ATM* that is involved in the repair of DNA damage after ionizing irradiation to be associated with the risk of brain tumors.[45] Mutations in DNA repair pathways are typically associated with therapeutic resistance and chemotherapy-induced mutagenesis.[46–48] This highlights the importance of genetic assessment following surgical resection. Should *RAD51C*- and *RNF43*-mutated pLGGs recur and undergo malignant progression to a higher histological grade, postoperative adjuvant treatment using immunotherapy approaches and checkpoint inhibitors may be employed as opposed to chemotherapeutic interventions. Regarding RNF43, mutations affecting this gene were loss-of function mutations, likely leading to the activation of pro-oncogenic Wnt signaling by interfering with the RNF43-mediated ubiquitination of the frizzled receptor.[49] Concurrent inhibition of WNT signaling components may therefore benefit these pLGG patients. In this regard, over 25 antibodies, 53 polypeptides/proteins and 21 chemicals are currently available to inhibit WNT signaling, some of which are clinically approved.[50,51]

Regarding our case report, the identification of the GOPC-ROS1 fusion provides insight into disease pathophysiology and the use of the FDA-approved ROS1 inhibitor crizotinib in SA pLGG because this was identified as a gain-of-function mutation in the Oncomine database (S2 Fig). Previous studies have also highlighted the *GOPC-ROS1* fusion as a resistance marker to chemotherapy in lung cancer, indicating that this fusion may be prognostic for a poor chemotherapeutic outcome.[41] As this patient achieved a gross total resection, there was no indication for adjuvant chemotherapy or radiation therapy. To date, the patient is stable with no symptoms or signs suggestive of tumor recurrence. Given the approval of crizotinib to treat late-stage non-small cell lung cancers that are ROS1 positive,[41,52] this may represent a potential treatment option should this patient experience disease recurrence.

## Conclusions

Interrogation of the sequencing data in the SA cohort has revealed *BRAF* fusions as critical biomarkers to predict resectable pLGG. We have further identified that *RAD51C*, *RNF43*, and *ATM* may hold prognostic value in the SA population. We identified a rare *GOPC-ROS1* fusion in pLGG patients lacking *BRAF* alterations, which may represent a genomically-distinct subgroup of pLGGs that could be targeted with crizotinib. To our knowledge this is the first report of this fusion in pLGG. These findings demonstrate how genetic profiling can guide optimal treatment strategies for pLGG in the Saudi population.

## Supporting information

**S1 Fig. Proposed pLGG testing strategy and diagnostic approach.**
(TIFF)

**S2 Fig. Gene targets in the Oncomine Assay in which identified pLGG mutants are highlighted.**
(TIFF)

**S1 Table. Complete patient demographics.**
(TIFF)

**S2 Table. Genetic landscape of LGG tumors.**
(TIFF)

## Acknowledgments

The authors thanks King Abdulaziz City for Science and Technology and the Saudi Human Genome Project for technical support. This Study was funded by KFMC-IRF 17–65 (MA) and Sanad Cancer research foundation RGP 2017–1 (MA)

## Author Contributions

**Conceptualization:** Lori A. Ramkissoon, Shakti H. Ramkissoon, Malak Abedalthagafi.

**Data curation:** Nahla A. Mobark, Musa Alharbi, Maqsood Ahmad, Ayman Al-Banyan, Fahad E. Alotabi, Duna Barakeh, Malak AlZahrani, Hisham Al-Khalidi, Abdulrazag Ajlan, Malak Abedalthagafi.

**Formal analysis:** Latifa AlMubarak, Rasha Alaljelaify, Mariam AlSaeed, Amal Almutairi, Fatmah Alqubaishi, Malak Abedalthagafi.

**Funding acquisition:** Malak Abedalthagafi.

**Investigation:** Nahla A. Mobark, Musa Alharbi, Malak Abedalthagafi.

**Methodology:** Nahla A. Mobark, Musa Alharbi, Malak Abedalthagafi.

**Project administration:** Malak Abedalthagafi.

**Resources:** Malak Abedalthagafi.

**Supervision:** Malak Abedalthagafi.

**Validation:** Malak Abedalthagafi.

**Writing – original draft:** Lamees Alhabeeb, Malak Abedalthagafi.

**Writing – review & editing:** Malak Abedalthagafi.

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
