## [Decision Letter · Decision Letter 0]

6 Nov 2019

PONE-D-19-25029

Clinical management and genomic profiling of pediatric low-grade gliomas in Saudi Arabia

PLOS ONE

Dear Dr. Abedalthagafi,

Thank you for submitting your manuscript to PLOS ONE. After careful consideration, we feel that it has merit but does not fully meet PLOS ONE’s publication criteria as it currently stands. Therefore, we invite you to submit a revised version of the manuscript that addresses the points raised during the review process.

We would appreciate receiving your revised manuscript by Dec 21 2019 11:59PM. To enhance the reproducibility of your results, we recommend that if applicable you deposit your laboratory protocols in protocols.io, where a protocol can be assigned its own identifier (DOI) such that it can be cited independently in the future. For instructions see: http://journals.plos.org/plosone/s/submission-guidelines#loc-laboratory-protocols

We look forward to receiving your revised manuscript.

Kind regards,

Jonathan H Sherman

Academic Editor

PLOS ONE

Journal Requirements:

2. In the ethics statement in the manuscript and in the online submission form, please provide additional information about the patient records and tissue samples used in your retrospective study, including: a) whether all data and tissue samples were fully anonymized before you accessed them; b) the date range (month and year) during which patients' medical records and tissue samples were accessed

3. Please include the full name of the IRB/ethics committee that reviewed and approved this study, including the name of the affiliated institution if applicable. We additionally ask that you include your IRB/ethics committee approval number in your ethics statement. Once you have amended this/these statement(s) in the Methods section of the manuscript, please add the same text to the “Ethics Statement” field of the submission form (via “Edit Submission”).

"The authors have declared that no competing interests exist.".

We note that one or more of the authors are employed by a commercial company: 'Foundation Medicine Inc'.

Reviewers' comments:

Reviewer's Responses to Questions

**Comments to the Author**

1. Is the manuscript technically sound, and do the data support the conclusions?

Reviewer #1: Yes

Reviewer #2: Yes

2. Has the statistical analysis been performed appropriately and rigorously? 

Reviewer #1: N/A

Reviewer #2: Yes

3. Have the authors made all data underlying the findings in their manuscript fully available?

Reviewer #1: No

Reviewer #2: Yes

4. Is the manuscript presented in an intelligible fashion and written in standard English?

Reviewer #1: Yes

Reviewer #2: Yes

5. Review Comments to the Author

Reviewer #1: In Mobark et al., the authors performed genetic profiling on 37 Saudia Arabian patients with Pediatric Low Grade Gliomas (pLGGs) in order to explore whether genetic mutations can correlate with histological and clinical features pLGGs to guide prognosis and therapeutics in their patient population. While this manuscript contributes data that has utility for the understanding of pLGGs, there are some points that the reviewer would like to see addressed.

Figures 1-3 summarize the clinical, histological, and genetic profiles of the patient base aptly. What is missing however is a display of the data that allows the reader to scan for potential relationships between demographics, clinical metrics, and tumor genotypes. Potential trends in correlations and, just as important, lack of correlations in the metrics are important to report to the literature.

The abstract indicates that an 8-year old patient lacking BRAF alterations underwent gross total resection followed, chemotherapy and hematopoietic stem cells, and remained disease free. In the results section (Lines 182-197), a 4 year old Pediatric High Grade Glioma (pHGG) patient with the GOPC-ROS1 fusion mutation received the same treatment and achieved the same outcome (previously reported in reference 39), whereas an 8-year old patient with the GOPC-ROS1 fusion mutation received a gross resection with no adjuvant chemotherapy. This passage, the abstract, or both need clarification.

It is inappropriate to describe the NGS-based genotyping approach taken in this study as a genomic approach. A targeted genetic screen of a panel of cancer-related genes does not constitute a genomic profile. The reviewer requests that the manuscript text is edited accordingly.

Reviewer #2: The authors reviewed a series of 37 pediatric low grade gliomas treated in their hospital in Saudi Arabia, and conducted genomic profiling of all the patients. They present their findings of the genetic profiling. They found a BRAF fusion was the most common alteration, and they present other gene alterations. They found one patient with a a GOPC-ROS1 mutation, and suggested possible clinical implications if the tumor recurs.

This paper surveyed the genetic landscape of pediatric low grade gliomas in their hospital, and suggested possible clinical implications. Here are some suggestions:

1. They distinguish cerebellar tumors from posterior fossa tumors in the figure and text. This is confusing since a cerebellar tumor is by definition a post fossa tumor. This should be clarified.

2. Figure 1 bar graphs of type of surgery, and relapse are unnecessary. This data could be provided in a table.

3. They described one patient whom they treated with radiation - it would be nice if they could give the rationale for radiation in that case.

4. They include a case of a patient with GOPC-ROS1 fusion pHGG. Why was this included in the series, if this series was about LGG? If they thought it was misclassified, why was it misclassified?

Interesting paper. I think it could be accepted with some revisions.

6. PLOS authors have the option to publish the peer review history of their article (what does this mean?). If published, this will include your full peer review and any attached files.

Reviewer #1: No

Reviewer #2: No

---

## [Author Response · Author response to Decision Letter 0]

6 Dec 2019

Reviewer #1: 

(1): Figures 1-3 summarize the clinical, histological, and genetic profiles of the patient base aptly. What is missing however is a display of the data that allows the reader to scan for potential relationships between demographics, clinical metrics, and tumor genotypes? Potential trends in correlations and, just as important, lack of correlations in the metrics are important to report to the literature.

Response: We thank you for this comment. The demographics and clinical metrics were descried in supplementary Table 1. 

(2) The abstract indicates that an 8-year old patient lacking BRAF alterations underwent gross total resection followed, chemotherapy and hematopoietic stem cells, and remained disease free. In the results section (Lines 182-197), a 4 year old Paediatric High Grade Glioma (pHGG) patient with the GOPC-ROS1 fusion mutation received the same treatment and achieved the same outcome (previously reported in reference 39), whereas an 8-year old patient with the GOPC-ROS1 fusion mutation received a gross resection with no adjuvant chemotherapy. This passage, the abstract, or both need clarification.

Response: We apologise for the confusion. The treatment regimens of our patient have been modified and corrected for clarity. We thank you for this comment.

 (3) It is inappropriate to describe the NGS-based genotyping approach taken in this study as a genomic approach. A targeted genetic screen of a panel of cancer-related genes does not constitute a genomic profile. The reviewer requests that the manuscript text is edited accordingly.

Response: We thank you for the comment. This has been modified throughout the manuscript as correctly requested.

Reviewer #2: 

1. They distinguish cerebellar tumors from posterior fossa tumors in the figure and text. This is confusing since a cerebellar tumor is by definition a post fossa tumor. This should be clarified.

Response: Thank you for the comment. This has now been combined as cerebellum/posterior fossa as correctly suggested.

2. Figure 1 bar graphs of type of surgery, and relapse are unnecessary. This data could be provided in a table.

These data have been shown as a Table in the modified Figure as requested.

3. They described one patient whom they treated with radiation - it would be nice if they could give the rationale for radiation in that case.

We thank you for the comment. Surgery did not provide complete resection so radiotherapy was employed this posterior fossa tumor. We have included this text to the manuscript for clarity.

4. They include a case of a patient with GOPC-ROS1 fusion pHGG. Why was this included in the series, if this series was about LGG? If they thought it was misclassified, why was it misclassified?

We apologise for the confusion. Our case was classified as pLGG . The pHGG discussions refer to the GOPC (FIG)-ROS1 fusion in a paediatric high-grade glioma survivor [reference 39 in the manuscript]. We have polished the text describing the treatment regimen of our pLGG patient for clarity.

---

## [Decision Letter · Decision Letter 1]

14 Jan 2020

Clinical management and genomic profiling of pediatric low-grade gliomas in Saudi Arabia

PONE-D-19-25029R1

Dear Dr. Abedalthagafi,

We are pleased to inform you that your manuscript has been judged scientifically suitable for publication and will be formally accepted for publication once it complies with all outstanding technical requirements.

With kind regards,

Jonathan H Sherman

Academic Editor

PLOS ONE

Additional Editor Comments (optional):

Reviewers' comments:

Reviewer's Responses to Questions

**Comments to the Author**

1. If the authors have adequately addressed your comments raised in a previous round of review and you feel that this manuscript is now acceptable for publication, you may indicate that here to bypass the “Comments to the Author” section, enter your conflict of interest statement in the “Confidential to Editor” section, and submit your "Accept" recommendation.

Reviewer #2: All comments have been addressed

2. Is the manuscript technically sound, and do the data support the conclusions?

Reviewer #2: (No Response)

3. Has the statistical analysis been performed appropriately and rigorously? 

Reviewer #2: (No Response)

4. Have the authors made all data underlying the findings in their manuscript fully available?

Reviewer #2: (No Response)

5. Is the manuscript presented in an intelligible fashion and written in standard English?

Reviewer #2: (No Response)

6. Review Comments to the Author

Reviewer #2: (No Response)

7. PLOS authors have the option to publish the peer review history of their article (what does this mean?). If published, this will include your full peer review and any attached files.

Reviewer #2: No

---

## [Editor Report · Acceptance letter]

21 Jan 2020

PONE-D-19-25029R1 

Clinical management and genomic profiling of pediatric low-grade gliomas in Saudi Arabia 

Dear Dr. Abedalthagafi:

I am pleased to inform you that your manuscript has been deemed suitable for publication in PLOS ONE. Congratulations! Your manuscript is now with our production department. 

With kind regards,

on behalf of

Dr. Jonathan H Sherman 

Academic Editor

PLOS ONE